# Social Capital as a Mediator through the Effect of Education on Depression and Obesity among the Elderly in China

**DOI:** 10.3390/ijerph17113977

**Published:** 2020-06-04

**Authors:** Yu Xin, Xiaohui Ren

**Affiliations:** West China School of Public Health and West China Fourth Hospital, Sichuan University, Chengdu 610041, China; xinyu10061006@163.com

**Keywords:** education, social capital, depression, obesity, older adults, mediator

## Abstract

Objectives: Global aging is an increasingly serious problem. The health problems faced by the elderly, such as depression and obesity, require serious consideration. Education, depression and obesity are inextricably linked; for the elderly, education is constant, and the factors which can mediate the relationship between education, depression and obesity are still being discussed by scholars. The mediating effect of social capital is rarely studied. The objective of this study was to assess the mediating role of cognitive social capital and structural social capital, as well as the effect of education on depression and obesity among the elderly using China Family Panel Studies (CFPS) data. Methods: In total, 4919 respondents were included in the final analysis. Education was measured by years of schooling. Trust and participation were used as measures of cognitive social capital and structural social capital. Depression symptoms and BMI were used as outcomes. Structural equation models were developed to examine the direct and indirect effect of social capital and education on health outcomes. Results: Education was negatively correlated with depression symptom (r = −0.15, *p* < 0.001), while education was positively correlated with BMI (r = 0.08, *p* < 0.001). Older adults with a higher education level have higher cognitive social capital (r = 0.11, *p* < 0.001) and structural social capital (r = 0.20, *p* < 0.001). Social capital plays a mediatory role. Older adults with higher social capital have a lower risk of depression (cognitive: r = −0.23, *p* < 0.001; structural: r = −0.03, *p* < 0.01) but a higher risk of obesity (cognitive: r = 0.06, *p* < 0.01; structural: r = 0.03, *p* < 0.01). For depression, the mediating function of cognitive social capital (a1b1= −0.025) is stronger than that of structural social capital (a2b2 = −0.006). While, for obesity, the effects of both cognitive and structural social capital are the same (a1c1 = a2c2 = 0.005). Conclusions: Social capital as a mediator through the effect of education on depression and obesity among the elderly in China. Meanwhile, using the positive effects of social capital to avoid negative effects should also be seriously considered.

## 1. Introduction

It is estimated that between 2017 and 2050, the proportion of the world’s population aged 60 years old and above will nearly double from 926 million in 2017 to 2.1 billion in 2050 [1]. In 2050, 80% of older people will be living in low- and middle-income countries. Aging is a challenge for all countries, and the crisis is even greater in low- and middle-income countries [2]. Older people are more likely to suffer from both mental and physical health problems [3,4,5]. Obesity and depression are particularly common health problems for the elderly. Over the past 40 years, the prevalence of global obesity has increased substantially, from less than 1% in 1975, to 6–8% in 2016 [6]. Furthermore, the prevalence of overweight and obesity increases with age [7]. Obesity is associated with daily living, chronic disease, and injuries among older people [8,9,10,11]. This is particularly concerning, given that obesity has seriously affects the health of the elderly. 

Globally, more than 264 million people of all ages suffer from depression [12]. They represent approximately 6.8% of the world’s population [13]. For the elderly, depression is a common psychological problem, which is characterized by a poor prognosis and periodical relapse [14]. It is also an important predictor of suicide in the elderly [15]. The health problems of obesity and depression among old adults in China are likely to exist in other developing countries as well [16,17]. 

Education is supposed to be a key influential factor shaping both mental and physical health. There is a large body of evidence highlighting huge disparities in health across subgroups of the elderly population, with individuals with poor educational attainment being disproportionately more likely to have depressive symptoms [16,17,18]. The relationship between education and obesity varies in different countries. In some developed countries, the higher the level of education, the lower the rate of obesity, but in many developing countries, the higher the level of education, the higher the rate of obesity/overweight [19,20]. Studies have shown that older people with lower cognitive function have higher rates of all-causes mortality. There is a bidirectional relationship from intelligence to health, through genetic factors and environment factor. Learning and reasoning are crucial in promoting health-protective behaviors, and education is likely to be a marker for these cognitive resources [21,22]. Fabrizio has studied the relationship between education and depression, mediated by health-related behaviors, working conditions and social engagement/personal control, but did not find any mediating effects [8]. Makambi found that vigorous physical activity mediates the association between education and obesity [23], but there are no studies showing that education through social capital affects obesity.

Pierre Bourdieu formally defined “social capital” by suggesting that it is the aggregate of the actual or potential resources which are linked to the possession of a durable network of more or less institutionalized relationships of mutual acquaintance and recognition—or, in other words, membership of a group [24]. Social capital can also be defined as features of social organization, such as trust, norms and civic participation [25].

Based on the theories put forward by former sociologists, social capital is split into structural social capital and cognitive social capital [26]. Cognitive social capital is derived from mental processes, which reflects the subjective evaluation of individuals’ bonds with others and encapsulates perceptions of the level of trust, confidence and shared values, norms and reciprocity, while the structural dimension refers to the observable aspects of social capital involving social networks and participation, as well as the properties of networks, relationships and institutions connecting individuals and groups [27]. In addition, another division of social capital is that between bonding and bridging: bonding social capital refers to the strong attachments that form between people who are similar to each other, such as within groups, and bridging social capital reflects weaker ties between people from different social backgrounds, such as a “between-group” relationship [28].At the same time, social capital can be divided into different levels. It can be measured as both an individual-level construct and as a group-level construct by aggregating individual perception to the community level [29]. In research into social capital, the classification of structural social capital and cognitive social capital is common.

Social capital is associated with depression [30]. Regarding cognitive social capital, people who can trust their friends, family and neighbors have a lower risk of depression [31]. Meanwhile, neighborhood trust has been shown to be a protective factor for depressive symptoms [32]. However, the effects of structured social capital on depression are inconsistent, with some studies suggesting it is not associated with depressive symptoms [33], and others suggesting it has a positive effect [34]. Regarding obesity, studies have shown that people with higher cognitive social capital (trust) and higher structural social capital (regular social participation) are less likely to be obese than those with low levels of trust and infrequent activity [35,36,37].

Although there is substantial evidence that social capital is a mediator between income and health [38,39,40,41], only a few studies have examined the role of social capital in the relationship between education and health [42,43]. A study in Latin America showed that trust at the country-level mediated the association between education and self-rated health, resulting in a higher trust level in a country, and higher odds of lower-educated individuals reporting good self-rated health [43]. However, other studies have found that social capital (e.g., social engagement) cannot play a mediating role between education and self-rated health [42]. Most of the above studies are from Western countries; it cannot be ignored that Eastern and Western cultures are different in many ways. Unlike the individualism of many Western countries, many Eastern countries advocate collectivism [44,45]. People in Eastern countries prioritize their family, friends and their groups over themselves and appreciate traditional values contributing to group solidarity and harmonious relationships among group members [46,47]. For example, Chinese people will pay more attention to other people’s opinions, leading to a greater potential to care about their weight. Such cultural differences between China and the West may affect the relationship between education, social capital and health. In addition, previous cross-sectional studies on social capital mediating education and health have used self-rated health as an outcome variable [42,43]. Hardly any studies have used physical indicators of health as an outcome. Body mass index (BMI) and depression symptoms could be considered as complementary measurements of health in older people, as they reflect physical and mental health.

Therefore, this study intended to discuss the mediating role of cognitive social capital and structural social capital through the effect of education on depression and obesity among the elderly, using China Family Panel Studies (CFPS) data.

We propose a theoretical framework in Figure 1. It shows the effect of cognitive social capital and structural social capital in the relationship between education and depression and obesity. Then thus, we propose the following hypotheses:

**Hypothesis** **1.**
*Higher education lowers the risk of depression, but increases the risk of obesity.*


**Hypothesis** **2.**
*The effect of education on depression and obesity is mediated by cognitive social capital and structural social capital.*


**Hypothesis** **3.**
*For depression, the mediating function of cognitive social capital is stronger than that of structural social capital.*


**Hypothesis** **4.**
*For obesity, the mediating function of cognitive social capital and structural social capital are same.*


## 2. Materials and Methods

The data were derived from the China Family Panel Studies (CFPS). CFPS is a biennial longitudinal survey which is conducted by the Institution of Social Science Survey at Peking University. This investigation launched in 2010, with five waves of publicly released datasets comprising the years 2010, 2012, 2014, 2016 and 2018. The samples covered 25 provinces, accounting for 95% of the total population of China. The contents of CFPS are rather typical, covering the demographics, socioeconomic condition, education and health of respondents. For the purpose of this paper, we used data from 2018 and selected a sample of respondents aged 60 or older.

The database included a national sample of 5087 people over the age of 60. We hoped that no missing data for all required variables. In this sample, we excluded respondents with missing explanatory variables, leaving 4919 respondents as the sample in this paper.

### 2.1. Dependent Variable

This paper used two indicators as dependent variables. The first was subjective health (depression) and the second was objective health (body mass index (BMI)).

In the 2018 CFPS questionnaire, depressive symptoms were measured by The Center for Epidemiologic Studies Depression Scale (CES-D) [48]. CFPS adopted a scale of eight questions, including six positive questions and two negative question. CES-D 8 measures the frequency of the following emotion in the past week: (1) I felt depressed, (2) I felt that everything was an efforts, (3) I slept restlessly, (4) I was happy, (5) I felt lonely, (6) I enjoyed life, (7) I felt sad, (8) I could not get going. This scale allows respondents to self-rate their degree of experience using a four-point scale: “rarely or never (less than 1 day)”, “not too often (1–2 days)”, “sometimes or half the time (3–4 days)”, and “most of the time (5–7 days)”. The responses for the items of negative feelings were assigned to an index value of 0, 1, 2 and 3, and those to the two positive feelings were assigned as 3, 2, 1 and 0. All scores were summed, with a range from 0 to 24. Depression as a continuous phenomenon with higher scores indicated higher depressive symptom levels.

BMI, a continuous measure of body weight relative to height, was calculated as body weight in kilograms divided by squared height in meters (kg/m^2^). The higher the BMI, the greater the risk of overweight and obesity.

### 2.2. Independent Variable

Education was measured by the number of years of education the respondents had completed by 2018.

### 2.3. Mediator

Mediating variables included two types of social capital: structural social capital and cognitive social capital.

The latent construct of cognitive social capital (CSC) was assessed by three trust indicators: (1) trust in parents, (2) trust in neighbor and (3) trust in stranger. Regarding trust, respondents were asked to rate how much they trusted each of the three groups on a 10-point scale, with 0 being very distrustful and 10 very trusting. Higher scores for each indicator meant higher levels of cognitive social capital.

The latent construct of structural social capital (SSC) was examined by one indicator: participation in labor unions. For structural social capital, the most important facets were the presence or the absence of network ties and network configuration. Participation in labor unions facilitates the accrual of obligation and favors group cohesion; at the same time, a labor union provides material and emotional support to its members. Participation in labor unions is a suitable indicator for measuring structural social capital. Respondents were asked whether they were members of a union (binary variable: 0 = no, 1 = yes). If the respondent joined labor unions, then this was scored as 1; if not, this was scored as 0. High scores indicated high structural social capital.

### 2.4. Other Socio-Demographics

Socio-demographics included sex, age, marital status and personal annual income. Age and annual income were continuous variables. Marital status was defined as a binary variable: “married” and “divorced, widowed, or unmarried”.

### 2.5. Data Analysis

This paper first calculated descriptive statistics (frequencies, percentage, means and standard deviations) to describe the sample. Subsequently, structural equation modeling (SEM) was used to examine the proposed hypotheses in IBM SPSS Statistics 21.0 (IBM, Armonk, NY, USA). Specifically, SEM was conducted in two steps: a measurement model and a structural model [49]. A classic estimator of SEM, maximum likelihood, is based on the assumption that observed variables are continuous and normally distributed [49,50,51]. A range of fit indexes were used, including the Chi-square test, the root mean square error of approximation (RMSEA), the comparative fit index (CFI), the Tucker–Lewis index (TLI) and the normed fit index (NFI) [49].

Due to the large sample size (*n* = 4919), it is not uncommon for a well-fitting hypothesized model to yield a significant χ2. The following criteria were used to determine a good model fit: RMSEA values lower than 0.05, and CFI, NFI and TLI values higher than 0.90. All statistical tests were two-sided (*p* = 0.05). Bootstrapping was used to test the statistical significance of the direct, indirect and total effects of the model.

## 3. Results

### 3.1. Descriptive Results

Table 1 presents the socio-demographic characteristics of respondents. The respondents’ average age was 68.03 ± 6.27 years old, the average years of education was 4.41 ± 4.62 years, and the average personal annual income was 22,336 ± 47,641.61 ¥. More than half of the respondents were male (51.5%), and 83.4% were married.

Table 2 shows the mean scores of the key variables for the entire sample. The mean scores for depression and BMI were 13.69 ± 4.43 and 23.03 ± 3.71. Regarding cognitive social capital indicators, the average score for trust in parent was 9.00 ± 1.86, trust in neighbors was 7.00 ± 2.20 and trust in strangers was 5.81 ± 2.26. The total average score in cognitive social capital was 18.00 ± 4.16. Regarding structural social capital, the total average score was 0.10 ± 0.27.

### 3.2. Structural Model

This study used SEM to test the theoretical assumptions. With the addition of socio-demographics (age, gender, marital status and personal annual income) as covariates, the arrow direction among the core variables in the SEM remained unchanged, and the corresponding coefficients did not change significantly, meaning that socio-demographics were not confounding factors. Figure 2 shows the model in which all paths were statistically significant, and the model had an adequate fit: χ2 = 122.328, df = 9, *p* < 0.001, NFI = 0.903, IFI = 0.909, CFI = 0.908, MSEA = 0.050.

### 3.3. Mediation Effect of Education on Depression through Cognitive Social Capital and Structural Social Capital

Education is a protective factor for depression (r = −0.15, *p* < 0.001), with a higher level of education showing a lower level of depression. Both structural social capital and cognitive social capital have negative effects on depression. Older adults who had higher cognitive social capital scores (r = −0.23, *p* < 0.001) and structural cognitive social capital scores (r = −0.03, *p* < 0.01) were less likely to be depressed.

Regarding the indirect path, education predicted a positive effect on cognitive social capital (r = 0.11, *p* < 0.001), as did structural social capital (r = 0.20, *p* < 0.001).

In mediation models (Table 3), this research showed specific indirect effects. The specific indirect effects were a1b1 = −0.025 (through CSC) and a2b2 = −0.006 (through SSC). With this examination, we have determined that CSC likely was an important mediator. The difference between CSC and SCS was statistically significant. In other words, education had an indirect effect on depression through the mediation of either cognitive social capital or structural social capital, but cognitive social capital played a major role.

### 3.4. Mediation Effect of Education on BMI through Cognitive Social Capital and Structural Social Capital

Education had an obviously negative effect on obesity (r = −0.08, *p* < 0.001), with a higher level of education correlating to a higher level of BMI. Both structural social capital and cognitive social capital have positive effects on obesity. Older adults who had higher cognitive social capital scores (r = −0.06, *p* < 0.01) and structural cognitive social capital (r = −0.03, *p* < 0.01) were more likely to have a higher BMI.

In mediation models (Table 4), this research showed specific indirect effects. The specific indirect effects were a1c1 = 0.005 (through CSC) and a2c2 = 0.005 (through SSC). With this examination, we determined that the indirect effects of CSC and SCS were the same, and that the difference was not statistically significant. In other words, education had an indirect effect on BMI through the mediation of either cognitive social capital or structural social capital, and both had the same effect.

## 4. Discussion

This study found that education have a negative effect on depression while education was positively correlated with obesity. According to the criteria of effect size from Funder, Funder, D.C. and Ozer, D.J. [52], the effect size is medium on depression and small on obesity. As previously assumed, the elderly with higher education are more likely to have a higher BMI in China. In terms of the relationship between education and depression, the results of our study are similar to those of other researchers who showed that higher education is associated with lower levels of depression [53,54]. The positive correlation between education and obesity is consistent with the findings in many developing countries [55,56].

Social capital plays a mediatory role. Older adults with higher social capital had a lower risk of depression but higher risk of obesity. Older adults with higher structural and cognitive social capital had a lower risk of depression. It could be highly educated people are less likely to be unemployed, and they are more likely to have richer social networks [57]. Kyu-Man Han’s research also finds that cognitive social capital is indeed a mediator on depression and education in community-living elderly, but his research does not consider structural social capital [28]. Cao found that cognitive social capital (trust and reciprocity) and structural social capital (social network) were negatively correlated with depression among Chinese people aged 60 years old and above [58]. However, we regard participation in labor unions as an indicator of structural social capital, which has urban–rural differences. This data show that nearly three-quarters of labor union participants are urban residents, so this metric is more inclined to explain the role of social capital for urban residents.

In line with our hypothesis, the mediatory effects of structural social capital and cognitive social capital on obesity and depression are different. For depression, cognitive social capital is a more important mediator, while for obesity, the effects of both are the same, which is consistent with the findings of other scholars [29,59]. For depression, the reason for this may be that cognitive social capital and depression symptoms both involve psychosocial processes and the perception of relationships [60]. Individuals who have a higher level of cognitive social capital tend to be more involved in reciprocal exchanges among family and neighbors and less likely to be exposed to social stress [61]. Structural social capital often relies on a political economy approach. Older people with higher structural social capital have better access to resources and are less vulnerable to financial concerns [60]; therefore, structured social capital can also reduce depression in old age.

Both structural and cognitive social capital had a mediating effect between obesity and education, and BMI was strongly enhanced by social capital in this study. However, other studies showed that cognitive and structural social capital were negatively correlated with BMI; in other words, people with higher social capital are less likely to be obese [36,37]. Obesity in China is mainly attributed to changes in diet and a decline in physical activity [60,62]. People with a higher education level tend to have lighter physical demands and a better economic condition. Obesity is a consequence of human behavior in the improvement of living standards in China [56]. High levels of education and high levels of social capital need to be built and sustained through behavior such as eating out; that is, highly-educated individuals have a higher possibility of eating food away from home [63], which is positively correlated to BMI [35,64]. It is surprising that elderly people over 60 years of age tend to eat food away from home more often [63]. This study had three limitations. First, it was cross-sectional, and therefore reverse causation cannot be ruled out. Longitudinal studies are needed to measure how education influences obesity and BMI through social capital. Second, there are different levels of social capital, and the mediating effect of each level may be different. This paper does not distinguish between the levels of social capital, and future studies should consider whether the mediating effect of different levels of social capital is different. Third, obesity, depression and social capital all have gender differences. Women are more obese in low- and middle-income countries, but the gender gap disappears in high-income economies [65]. Meanwhile, previous studies found that women report higher levels of depression than men do [66]. Social capital may also have gender differences [67]. Therefore, gender differences may exist in the mediating role of social capital, which is also the content of our future research.

## 5. Conclusions

Social capital as a mediator through the effect of education on depression and obesity among the elderly in China. For depression, social capital was negatively correlated with depression, and cognitive social capital could play a major role. For obesity, social capital was positively correlated with BMI, and structural social capital could play a similar role to cognitive social capital. The finding could have some important implications for health intervention on the elderly in China. Education is a constant for the elderly, and we could use social capital (cognitive social capital and structural social capital) to adjust its relationship with health. Notably, social capital could have a two-sided effect. We should try to avoid the negative influence. Meanwhile, our study enriched the research on the mediating role of social capital, and put forward suggestions for developing countries to fully consider the two-side effect on social capital. In future research, we can verify the mediating role of social capital through cohort study, and further consider the impact of urban and rural areas, gender and other aspects.

## Figures and Tables

**Figure 1 ijerph-17-03977-f001:**
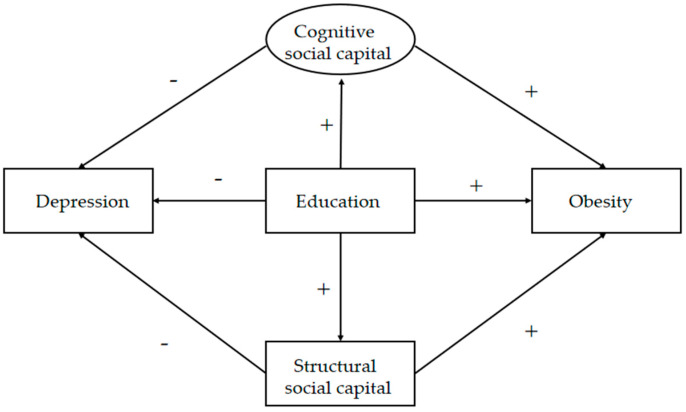
Conceptual framework.

**Figure 2 ijerph-17-03977-f002:**
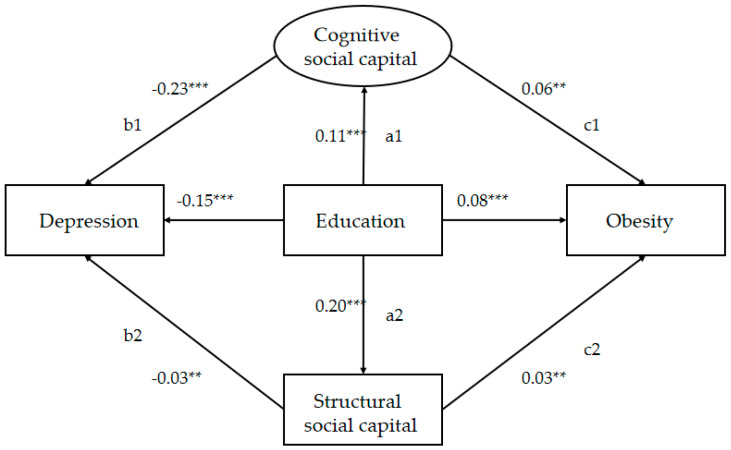
Solution for the structural model ** *p* < 0.01 *** *p* < 0.001.

**Table 1 ijerph-17-03977-t001:** Characteristics of the study sample (*n* = 4919).

Variable	*n* (%)	M (SD)
Age	-	68.03 (6.27)
Average years of education	-	4.41 (4.62)
Personal annual income (¥)	-	22,336 (47,641.61)
Gender		
Male	2525 (51.5)	
Female	2384 (48.5)	
Marital status		
Married	4101 (83.4)	
Divorce/Separate/Widowed	818 (16.6)	

**Table 2 ijerph-17-03977-t002:** Mean scores of key variables for the total sample.

Variable	M (SD)
Depression	13.69 (4.43)
BMI	23.03 (3.71)
Cognitive social capital	18.00 (4.16)
Trust in parents	9.00 (1.86)
Trust in neighbor	7.00 (2.20)
Trust in stranger	1.99 (2.26)
Structural social capital	0.10 (0.27)

**Table 3 ijerph-17-03977-t003:** Mediation of effect of education on depression through cognitive social capital and structural social capital. EDU: education; CSC: cognitive social capital; DEP: depression; SSC: structural social capital.

Model	Point Estimation	Product of Coefficients		Bootstrapping			
				Bias-Corrected 95% CI		Percentile 95% CI	
		SE	Z	Lower	Upper	Lower	Upper
EDU→CSC→DEP	−0.025 **	0.006	4.167	−0.035	−0.012	−0.034	−0.011
EDU→SSC→DEP	−0.006 *	0.002	3.000	−0.011	−0.002	−0.011	−0.002
Different comparison	−0.018 *	0.007	2.571	−0.031	−0.004	−0.030	−0.004

Note: Standardized estimating of 1000 bootstrap sample, * *p* < 0.1, ** *p* < 0.05.

**Table 4 ijerph-17-03977-t004:** The mediation of effect of education on obesity through cognitive social capital and structural social capital.

Model	Point Estimation	Product of Coefficients		Bootstrapping			
				Bias-Corrected 95% CI		Percentile 95%CI	
		SE	Z	Lower	Upper	Lower	Upper
EDU→CSC→BMI	0.005 **	0.002	2.500	0.001	0.010	0.001	0.010
EDU→SSC→BMI	0.005 **	0.002	2.500	0.001	0.009	0.001	0.009
Different comparison	0.000	0.003	0.000	−0.006	0.007	−0.006	0.007

Note: Standardized estimating of 1000 bootstrap sample, ** *p* < 0.05.

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
