# Peer review of "Social Capital as a Mediator through the Effect of Education on Depression and Obesity among the Elderly in China"

_ijerph, 2020, doi:10.3390/ijerph17113977_

Round 1

Reviewer 1 Report

The paper assess the mediating role of cognitive social capital and structural social capital on education, depression and obesity among the elderly peolple in China. This is a solid paper devoted to the pertinent issue of social capital and nexus between education and health.

The paper stands out in its methodological argumentation, which makes the results understandable and relatable.

This research relies heavily on an array of key concepts which hang loose and beg for questions:

  • The multidimensional and composite nature of capital social makes any attempt to understand and measure difficult. For this reason it is necessary to start from the identification and description of the various dimensions that make it up.
  • I need more explanation on the choice of the indicator participation in labor unions as the latent construct of structural social capital. The cognitive dimension of social capital includes those elements of social organization that allow members of a community to reach a shared representation of their own community. Among these elements, noteworthy is covered with values and beliefs elements that characterize local culture (Fukuyama, 1995, 2000). For this reason, I ask the authors if it was not possible to use other "cultural" indicators to emphasize the value system (and measure) of a community. I understand that the choice of labor union can be an indicator of compatibility of the values internalized by the individual with those of the community, but perhaps I would prefer that the concept be better explained in the paper.
  • The paper does not consider the role of the territory. Perhaps it can be useful, at least in the discussion of the results, to try to provide some reflection (for example on the difference of the results between rural and urban areas). This is very important to broaden the reflection on the role of social capital as a determinant of local development and to broaden the international dimension of the paper.

Finally, there are serval minor issues for considering:

  • In the title I would replace BMI with obesity.
  • In order to help readers I suggest to use acronyms only after entering the words to which they refer (i.e. BMI in the title, WHO line 34).

Author Response

Point 1: The multidimensional and composite nature of capital social makes any attempt to understand and measure difficult. For this reason it is necessary to start from the identification and description of the various dimensions that make it up.

Response 1: We have added more details about social capital (Line 63-76). First, we have improved the definition of structural social capital and cognitive social capital. Cognitive social capital is derived from mental processes, which reflects the subjective evaluation of individuals’ bonds with others and encapsulates perceptions of the level of trust, confidence, shared values, norms and reciprocity, while the structural dimension refers to the observable aspects of social capital involving social networks and participation, as well as the properties of networks, relationships and institutions connecting individuals and groups. Second, we have added other classification standards of social capital and described them (bonding and bridging). Bonding social capital refers to the strong attachments that form between people who are similar to each other, such as within groups, and bridging social capital reflects weaker ties between people from different social backgrounds, such as a “between-group” relationship. Third, we describe the level of social capital. It can be measured as both an individual-level construct and as a group-level construct by aggregating individual perception to the community level.

Point 2: I need more explanation on the choice of the indicator participation in labor unions as the latent construct of structural social capital. The cognitive dimension of social capital includes those elements of social organization that allow members of a community to reach a shared representation of their own community. Among these elements, noteworthy is covered with values and beliefs elements that characterize local culture (Fukuyama, 1995, 2000). For this reason, I ask the authors if it was not possible to use other "cultural" indicators to emphasize the value system (and measure) of a community. I understand that the choice of labor union can be an indicator of compatibility of the values internalized by the individual with those of the community, but perhaps I would prefer that the concept be better explained in the paper.

Response 2: We add the reason that labor unions are suitable latent indicator to social capital (Line 165-169). For structural social capital, the most important facets were the presence or the absence of network ties and network configuration. Participation in labor unions facilitates the accrual of obligation and favors group cohesion; at the same time, a labor union provides material and emotional support to its members. Participation in labor unions is a suitable indicator for measuring structural social capital. The original meaning of China's labor unions refers to a social organization spontaneously organized on the basis of common interests. In addition, joining an enterprise represents the identification of the enterprise culture. So, unionists share the same values and believe.

Point 3: The paper does not consider the role of the territory. Perhaps it can be useful, at least in the discussion of the results, to try to provide some reflection (for example on the difference of the results between rural and urban areas). This is very important to broaden the reflection on the role of social capital as a determinant of local development and to broaden the international dimension of the paper.

Response 3: We have added explanations of the territory (Line173-176). We regard participation in labor unions as an indicator of structural social capital, which has urban–rural differences. This data show that nearly three quarters of labor union participants are urban residents, so this metric is more inclined to explain the role of social capital for urban residents.

Point 4: In the title I would replace BMI with obesity.

Response 4: We also think that replacing BMI with obesity will make the article better, so we have changed the title to: Social Capital as a Mediator through the Effect of Education on Depression and Obesity among the Elderly in China.

Point 5: In order to help readers I suggest to use acronyms only after entering the words to which they refer (i.e. BMI in the title, WHO line 34).

Response 5: We have change “WHO” into “World Health Organization” (line 37).

Reviewer 2 Report

The article is interesting and might have educational and practical value. The topic of this paper is relevant and timely. The general method, participant, setting and measurement procedures are well described. However, before the article was accepted, a few modifications should be made. My minor comments are listed below:

BACKGROUND
Introduction section it little bite too general. Moreover I would suggest to explain more detailed West-East culture differences – why researches in Asia/China are so unique. May by those papers could be useful:

Różycka-Tran, J., Ha, T. T. K., Cieciuch, J., & Schwartz, S. H. (2017). Universals and specifics of the structure and hierarchy of basic human values in Vietnam. Health Psychology Report, 5(3), 193-204. https://doi.org/10.5114/hpr.2017.65857

Lipowska, M., Truong Thi Khanh, H., Lipowski, M., Różycka-Tran, J., Bidzan, M., & Ha, T. T. (2019). The Body as an Object of Stigmatization in Cultures of Guilt and Shame: A Polish–Vietnamese Comparison. International Journal of Environmental Research and Public Health, 16(16), 2814. https://doi.org/10.3390/ijerph16162814

Izydorczyk, B., Truong Thi Khanh, H., Lizińczyk, S., Sitnik-Warchulska, K., Lipowska, M. & Gulbicka, A. (2020). Body dissatisfaction, restrictive, and bulimic behaviours among young women: a Polish – Japanese comparison. Nutrients, 12(3), 666. https://doi.org/10.3390/nu12030666

METHODOLOGY

I would suggest to report the research hypothesis, they would add value to the study respecting the repeatability of the research

RESULTS

Statistical analysis and the presentation of results do not raise objections.

DISCUSSION

Discussions should integrate the themes suggested in the introduction section of this review. In general, they report the main points and finding of this research. I would suggest to structure this part with connection with hypotheses.

Moreover very interesting results connected with gender differences are discussed slightly, more specific references would be useful:

Lipowska, M., Lipowski, M., Olszewski, H., & Dykalska-Bieck, D. (2016). Gender differences in body-esteem among seniors: Beauty and health considerations. Archives of Gerontology and Geriatrics, 67, 160–170. https://doi.org/10.1016/j.archger.2016.08.006

Mohseni, M., Iranpour, A., Naghibzadeh-Tahami, A., Kazazi, L., & Borhaninejad, V. (2019). The relationship between meaning in life and resilience in older adults: a cross-sectional study. Health Psychology Report, 7(2), 133-138. doi:10.5114/hpr.2019.85659

Obara-Gołębiowska, M., Brycz, H., Lipowska, M., & Lipowski, M. (2018). The role of motivation to reduce obesity among elderly people: response to priming temptation in obese individuals. International Journal of Environmental Research and Public Health, 15(2), 244; https://doi.org/10.3390/ijerph15020244

Kroemeke, A., & Gruszczyńska, E. (2014). Depressive symptom clusters among the elderly: a longitudinal study of course and its correlates. Health Psychology Report, 2(4), 269-279. doi:10.5114/hpr.2014.46694

In summary, I recommend the text to be published in the International Journal of Environmental Research and Public Health after changes suggested by reviewer.

Author Response

Point 1: Introduction section it little bite too general. Moreover I would suggest to explain more detailed West-East culture differences – why researches in Asia/China are so unique. May by those papers could be useful:

Różycka-Tran, J., Ha, T. T. K., Cieciuch, J., & Schwartz, S. H. (2017). Universals and specifics of the structure and hierarchy of basic human values in Vietnam. Health Psychology Report, 5(3), 193-204. https://doi.org/10.5114/hpr.2017.65857

Lipowska, M., Truong Thi Khanh, H., Lipowski, M., Różycka-Tran, J., Bidzan, M., & Ha, T. T. (2019). The Body as an Object of Stigmatization in Cultures of Guilt and Shame: A Polish–Vietnamese Comparison. International Journal of Environmental Research and Public Health, 16(16), 2814. https://doi.org/10.3390/ijerph16162814

Izydorczyk, B., Truong Thi Khanh, H., Lizińczyk, S., Sitnik-Warchulska, K., Lipowska, M. & Gulbicka, A. (2020). Body dissatisfaction, restrictive, and bulimic behaviours among young women: a Polish – Japanese comparison. Nutrients, 12(3), 666. https://doi.org/10.3390/nu12030666

Response 1: In the introduction, we have added a detailed description of social capital and a more comprehensive literature to prove the hypothesis to be correct.

we explain the cultural differences between Eastern and Western cultures, from the perspective of collectivism and individualism (Line 91-98). Unlike the individualism of many Western countries, many Eastern countries advocate collectivism. People in Eastern countries prioritize their family, friends and their groups over themselves and appreciate traditional values contributing to group solidarity and harmonious relationships among group members. For example, Chinese people will pay more attention to other people's opinions, leading to a greater potential to care about their weight. Such cultural differences between China and the West may affect the relationship between education, social capital and health.

Point 2: I would suggest to report the research hypothesis, they would add value to the study respecting the repeatability of the research

Response 2: We add the hypothesis in the paper (Line 98-115). We also propose a theoretical framework in Figure 1. It shows the effect of cognitive social capital and structural social capital in the relationship between education and depression and obesity. Thus, we propose the following hypotheses:

Hypothesis 1. Higher education lowers the risk of depression, but increases the risk of obesity;

Hypothesis 2. The effect of education on depression and obesity is mediated by cognitive social capital and structural social capital;

Hypothesis 3. The mediation effect of cognitive social capital and structural social capital is different.

Fig.1.The conceptual framework.

Point 3: Statistical analysis and the presentation of results do not raise objections.

Response 3: We did not modify the results.

Point 4: Discussions should integrate the themes suggested in the introduction section of this review. In general, they report the main points and finding of this research. I would suggest to structure this part with connection with hypotheses.

Moreover very interesting results connected with gender differences are discussed slightly, more specific references would be useful:

Lipowska, M., Lipowski, M., Olszewski, H., & Dykalska-Bieck, D. (2016). Gender differences in body-esteem among seniors: Beauty and health considerations. Archives of Gerontology and Geriatrics, 67, 160–170. https://doi.org/10.1016/j.archger.2016.08.006

Mohseni, M., Iranpour, A., Naghibzadeh-Tahami, A., Kazazi, L., & Borhaninejad, V. (2019). The relationship between meaning in life and resilience in older adults: a cross-sectional study. Health Psychology Report, 7(2), 133-138. doi:10.5114/hpr.2019.85659

Obara-Gołębiowska, M., Brycz, H., Lipowska, M., & Lipowski, M. (2018). The role of motivation to reduce obesity among elderly people: response to priming temptation in obese individuals. International Journal of Environmental Research and Public Health, 15(2), 244; https://doi.org/10.3390/ijerph15020244

Kroemeke, A., & Gruszczyńska, E. (2014). Depressive symptom clusters among the elderly: a longitudinal study of course and its correlates. Health Psychology Report, 2(4), 269-279. doi:10.5114/hpr.2014.46694

Response 4: The influence of gender on the mediating role of social capital is the inadequacy of our study, and it is also a question that we need to focus on in the future. We have added this problem to the limitations of the discussion (Line 303-307). Obesity, depression and social capital all have gender differences. Women are more obese in low- and middle-income countries, but the gender gap disappears in high-income economies. Meanwhile, previous studies found that women report higher levels of depression than men do. Social capital may also have gender differences.

Reviewer 3 Report

IMPROVE THE QUALITY OF THE WRITING AND THE ENGLISH

Already the title and the first lines of Abstract make clear that this paper is not written by native speakers of English and that it requires a lot of improvement.

This paper should have definitely been checked by a native speaker of English, as is clearly required in the Notes for Authors of almost all international journals. Why did the authors not have the quality of the English checked before submitting their paper? That should be standard procedure. Now they force the reviewers to wrestle through all the low-quality English. The authors of the paper should make an effort, not the reviewer.

I will review a paper written in proper English, that is checked by a native speaker of English.

TITLE

Quite unclear, requires improvement.

ABSTRACT

L 9 “The aging is now sweeping the world.” Very unclear writing, not sure what the authors are trying to say.

‘education is unchangeable’ Please write correct English.

‘seldom be discussed’ Write correct English

Many more bad sentences.

Make the Research Question more explicit.

INTRODUCTION

  1. 33 ‘among of them’; Write correct English

Author Response

Point 1: I will review a paper written in proper English, that is checked by a native speaker of English.

Response 1: We gave our paper to MDPI Author Services. English Editing Services helps us to check our grammar and writing style.

Point 2: TITLE:Quite unclear, requires improvement.

Response 2: We have changed the title to: Social Capital as a Mediator through the Effect of Education on Depression and Obesity among the Elderly in China.

Point 3: ABSTRACT

L 9 “The aging is now sweeping the world.” Very unclear writing, not sure what the authors are trying to say.

‘education is unchangeable’ Please write correct English.

‘seldom be discussed’ Write correct English

Many more bad sentences.

Make the Research Question more explicit.

Response 3: We deleted the sentence: “The aging is now sweeping the world”; and change “education is unchangeable” into “education is constant”; “seldom be discussed’ into “is rarely studied”.

Point 4: INTRODUCTION

33 ‘among of them’; Write correct EnglishKroemeke, A., & Gruszczyńska, E. (2014). Depressive symptom clusters among the elderly: a longitudinal study of course and its correlates. Health Psychology Report, 2(4), 269-279. doi:10.5114/hpr.2014.46694

Response 4: We deleted “among of them”.

Round 2

Reviewer 1 Report

The authors did a major revision work that helped improve the previous version of the paper.

Author Response

Thank you for your advice.

Reviewer 3 Report

The quality of the writing and the English has improved dramatically. This is what the first version of the paper should have looked like. Mind you, I still see typos in the text, so it is clear that the authors did not use a standard spelling checker before submitting the new version of their manuscript.

This was a classic case of the low-quality of the English distracting from what is actually a quite good scientific paper.

It is good to have a paper focusing on the health of the elderly.

Good size research sample.

DISCUSS COGNITIVE EPIDEMICS

Education has been linked to health. However, what are the causal factors? It is well known that different levels of education reflect different levels of mean IQ. There is a whole field of science called cognitive epidemics, focusing on the link between IQ and health.  Think of the excellent work of Linda Gottfredson and Ian Deary. Discuss this work in detail, and see how it changes your conclusions.

For instance, Linda Gottfredson argues that it is a difficult task to stay healthy. Dealing with illnesses can be difficult, for instance how to deal with diabetes; many difficult choices are required, and lower-IQ people simply make a lot of bad decisions, because they do not understand the diabetes-related information well. Or think of people who have to keep a diet and have to judge the protein and carbohydrate content of food when they do not understand these concepts. Lower IQ is genetically determined, and there is a genetic component in lower-IQ people having lower health.

Lower education is partially caused by having lower IQ, and lower IQ is partially genetically determined.

ABSTRACT

Make the Research Question more specific. Not just studying the role of, but, based on theory and empirical findings, predict what the size of the effect is you expect. At the very least: positive effect or negative effect.

In Abstract it is not specified how the variables were analyzed.

In the equivalent of Results it is not specified how strong the effect were. We learn there is a correlation, but is it r = .01 or is it r = .99? Make the Abstract more informative.

There is not a clear section on what the outcomes of the statistical analyses mean, how to interpret the statistical outcome in light of the theories formulated in Introduction.

INTRODUCTION

The authors cite data from the WHO; in how far can WHO data be trusted? The recent Corona outbreak shows the incompetence of the leadership. Does the incompetence of the leadership lead to incompetence among the researchers? Maybe add some other sources or more strongly: do not cite data from the WHO.

  1. 48 chronic course? What is meant?

Make the Hypotheses more specific. What kind of effects do you expect? How strong will the correlations or effect sizes be?

Hypotheses 3 is very unclear. What is meant with ‘different’? Be precise in the formulation of your hypotheses.

METHOD

Great dataset.

  1. 138 You are talking about provinces of China, but Taiwan is an independent country. Keep political propaganda out of scientific papers.

Are there also intelligence data in the database? IQ could be used as an additional variable for the statistical analyses, for instance a control variable. See point on cognitive epidemics above.

Add some more details on what was actually done, which model was actually tested and why.

For the rest, good Method section.

DISCUSSION

  1. 273: Please specify the sizes of the relationships: miniscule, small, medium, large, very large.

People with higher education are more obese. This goes strongly against the findings from scientific studies in Western countries, which show the opposite. Use the brilliant paper by Funder & Ozer (2019):

Funder, D. C., & Ozer, D. J. (2019). Evaluating effect sizes in psychological research: Sense and nonsense. Advances in Methods and Practices in Psychological Science, 2, 156-168.

The Conclusions could be improved. Be more explicit what the theoretical and practical implications of the present findings are. Also be more explicit in suggestions for follow-up research.

Author Response

Point 1:

DISCUSS COGNITIVE EPIDEMICS

Education has been linked to health. However, what are the causal factors? It is well known that different levels of education reflect different levels of mean IQ. There is a whole field of science called cognitive epidemics, focusing on the link between IQ and health.  Think of the excellent work of Linda Gottfredson and Ian Deary. Discuss this work in detail, and see how it changes your conclusions.

For instance, Linda Gottfredson argues that it is a difficult task to stay healthy. Dealing with illnesses can be difficult, for instance how to deal with diabetes; many difficult choices are required, and lower-IQ people simply make a lot of bad decisions, because they do not understand the diabetes-related information well. Or think of people who have to keep a diet and have to judge the protein and carbohydrate content of food when they do not understand these concepts. Lower IQ is genetically determined, and there is a genetic component in lower-IQ people having lower health.

Lower education is partially caused by having lower IQ, and lower IQ is partially genetically determined.

Response 1: We also agree that there is a strong link between cognitive level and education. We have carefully read the relevant literature of Linda Gottfredson and Ian Deary then supplemented our introduction(Line 52-55). Studies have shown that older people with lower cognitive function have higher rates of all-causes mortality. There is a bidirectional relationship from intelligence to health, through genetic factors and environment factor. Learning and reasoning are crucial in promoting health-protective behaviors, and education is likely to be a marker for these cognitive resources.

The above contents are based on the following two literatures:

Rosalind Arden, Linda S. Gottfredson, Geoffrey Miller,Does a fitness factor contribute to the association between intelligence and health outcomes? Evidence from medical abnormality counts among 3654 US Veterans, Intelligence. 2009,37(6), 581-591.

https://doi.org/10.1016/j.intell.2009.03.008.

  1. David Batty, Ian J. Deary, Linda S. Gottfredson.Premorbid (early life) IQ and Later Mortality Risk: Systematic Review, Annals of Epidemiology.2007, 17(4), 278-288.

https://doi.org/10.1016/j.annepidem.2006.07.010.

Point 2:

ABSTRACT

Make the Research Question more specific. Not just studying the role of, but, based on theory and empirical findings, predict what the size of the effect is you expect. At the very least: positive effect or negative effect.

In Abstract it is not specified how the variables were analyzed.

In the equivalent of Results it is not specified how strong the effect were. We learn there is a correlation, but is it r = .01 or is it r = .99? Make the Abstract more informative.(

There is not a clear section on what the outcomes of the statistical analyses mean, how to interpret the statistical outcome in light of the theories formulated in Introduction.

Response 2:

We combed and supplemented our summary according to objective, method, result and conclusion (Line 10-31).

Abstract:  

Objectives: Global aging is an increasingly serious problem. The health problems of the elderly, such as depression and obesity, require serious consideration. Education, depression and obesity are inextricably linked; for the elderly, education is constant, and the factors which can mediate the relationship between education, depression and obesity are still being discussed by scholars. The mediating effect of social capital is rarely studied. The objective of this study was to assess the mediating role of cognitive social capital and structural social capital, as well as the effect of education on depression and obesity among the elderly using China Family Panel Studies (CFPS)data.

Methods: In total, 4919 respondents were included in the final analysis. Education was measured by years of schooling. Trust and participation were used as measures of cognitive social capital and structural social capital. Depression symptoms and BMI were used as outcomes. Structural Equation Model were developed to examine the direct and indirect effect of social capital and education on health outcomes.

 Results: Education was negatively correlated with depression symptom (r=-0.15, P<0.001), while education was positively correlated with BMI (r=0.08, P<0.001). Older adults with a higher education level have higher cognitive social capital (r =0.11, P<0.001) and structural social capital (r=0.20, P<0.001). Social capital plays a mediatory role. Older adults with higher social capital have a lower risk of depression (cognitive: r=-0.23, P<0.001; structural: r=-0.03, P<0.01) but a higher risk of obesity (cognitive: r=0.06, P<0.01; structural: r=0.03, P<0.01). For depression, the mediating function of cognitive social capital (a1b1= -0.025) is stronger than that of structural social capital (a2b2=-0.006). while for obesity, the effects of both cognitive and structural social capital are the same(a1c1=a2c2=0.005).

 Conclusions: Social capital as a mediator through the effect of education on depression and obesity among the elderly in China. Meanwhile, using the positive effects of social capital to avoid negative effects should also be seriously considered.

Point 3:

INTRODUCTION

The authors cite data from the WHO; in how far can WHO data be trusted? The recent Corona outbreak shows the incompetence of the leadership. Does the incompetence of the leadership lead to incompetence among the researchers? Maybe add some other sources or more strongly: do not cite data from the WHO.

Response 3:

We have added data on global aging, obesity and depression.

Global aging: We have added data from the United Nations population division(Line 36).

It is estimated that between 2017 and 2050, the proportion of the world's population aged 60 years old and above will nearly double from 926 million in 2017 to 2.1 billion in 2050

The literature cited is as follows:

United Nations (UN), Department of Economic and Social Affairs, Population Division. World Population Aging 2015 (ST/ESA/SER.A/390) Available from:

(2015) https://www.un.org/en/development/desa/population/publications/ageing/WPA2015_Infochart.asp

Obesity: we used the data from The Lancet (Line 40-41). Over the past 40 years, the prevalence of global obesity has increased substantially, from less than 1% in 1975, to 6–8% in 2016.

The literature cited is as follows:

Lindsay M Jaacks, Stefanie Vandevijvere, An Pan, Craig J McGowan, Chelsea Wallace, Fumiaki Imamura, Dariush Mozaffarian, Boyd Swinburn, Majid Ezzati. The obesity transition: stages of the global epidemic. The Lancet Diabetes & Endocrinology. 2019, 7(3), 231-240.

Depression: we have added the data from World Happiness Report (Line 46). Approximately 6.8% of the world’s population suffer from depression.

The literature cited is as follows:

Layard, R., Chisholm, D., Patel, V., Saxena, S., 2013. Mental illness and unhappiness. In:

Helliwell, J., Layard, R., Sachs, J. (Eds.), World Happiness Report 2013, pp. 38–53.

Point 4: 48 chronic course? What is meant?

Response 4: We changed “chronic course” into “periodical relapse”

Point 5: Make the Hypotheses more specific. What kind of effects do you expect? How strong will the correlations or effect sizes be?

Hypotheses 3 is very unclear. What is meant with ‘different’? Be precise in the formulation of your hypotheses.

Response 5: We described the hypothesis in more detail (Line 120-123).

Hypothesis 3. For depression, the mediating function of cognitive social capital is stronger than that of structural social capital;

Hypothesis 4. For obesity,the mediating function of cognitive social capital and structural social capital are same.

Point 6:

METHOD

138 You are talking about provinces of China, but Taiwan is an independent country. Keep political propaganda out of scientific papers.

Response 6: We deleted “not including Hong Kong, Macau and Taiwan”. However, Taiwan is a part of China.

Point 7:

Are there also intelligence data in the database? IQ could be used as an additional variable for the statistical analyses, for instance a control variable. See point on cognitive epidemics above.

Add some more details on what was actually done, which model was actually tested and why.

Response 7:

 (1) We agree that IQ has an impact, but unfortunately, the database we use does not contain indicators of intelligence. In our future education related research, we will consider intelligence as a control variable.

(2) We separately tested the structural equation model of depressive symptoms and BMI as results, and through extensive literature reading, especially Aida's paper, which makes us want to consider both psychological and physical health outcomes. In the test of the model, we based on the good fit (Line 183-185), and we also take demographic characteristics into account (Line 206). The following criteria were used to determine a good model fit: RMSEA values lower than0.05, and CFI, NFI and TLI values higher than 0.90. All statistical tests were two-sided (P = 0.05). Bootstrapping was used to test the statistical significance of the direct, indirect, and total effects of the model.

The literature as follows:

Aida, J.; Kondo, K.; Kondo, N.; Watt, R. G.; Sheiham, A.; Tsakos, G. Income inequality, social capital and self-rated health and dental status in older Japanese. Social Science & Medicine. 2011, 73, 1561-1568.

Point 8:

DISCUSSION

273: Please specify the sizes of the relationships: miniscule, small, medium, large, very large.

People with higher education are more obese. This goes strongly against the findings from scientific studies in Western countries, which show the opposite. Use the brilliant paper by Funder & Ozer (2019):

Funder, D. C., & Ozer, D. J. (2019). Evaluating effect sizes in psychological research: Sense and nonsense. Advances in Methods and Practices in Psychological Science, 2, 156-168.

Response 8:

  • We have modified the discussion according to the literature (Line 246-248). Education have a negative effect on depression while education was positively correlated with obesity. According to the criteria of effect size from Funder, D. C., & Ozer, D. J, the effect size is medium on depression and small on obesity.
  • People with higher education are more obese. This goes strongly against the findings from scientific studies in Western countries, which show the opposite. We did not find a relative study on Funder & Ozer (2019).

The literature we cited is as follows:

Stella T. Lartey.; Lei Si.; Petr Otahal.; Barbara de Graaff.; Godfred O. Boateng.; Richard Berko Biritwum et al. Annual transition probabilities of overweight and obesity in older adults: Evidence from World Health Organization Study on global AGEing and adult health. Social Science & Medicine .2020, 274, e112821.

Sulander, T. T.; A. K. Uutela. Obesity and education: Recent trends and disparities among 65-to 84-year-old men and women in Finland. Preventive Medicine. 2007, 45, 153-156.

Kinge, J. M.; Strand, B. H.; Vollset, S. E.; Skirbekk, V. Educational inequalities in obesity and gross domestic product: evidence from 70 countries. Journal of Epidemiology and Community Health. 2015, 69, 1141-1146.

Zhou, M. The shifting burden of obesity: Changes in the distribution of obesity in China, 2010-2015. International Sociology. 2019, 34, 347-367.

Point 9: The Conclusions could be improved. Be more explicit what the theoretical and practical implications of the present findings are. Also be more explicit in suggestions for follow-up research.

Response 9: We revised our conclusion section (Line 304-307). The finding could have some important implications for health intervention on the elderly in China. Education is a constant for the elderly, and we could use social capital (cognitive social capital and structural social capital) to adjust its relationship with health. Noteworthily, social capital could have a two-sided effect. We should try to avoid the negative influence. Meanwhile, our study enriched the research on the mediating role of social capital, and put forward suggestions for developing countries to fully consider the two-side effect on social capital. In the future research, we can verify the mediating role of social capital through cohort study, and further consider the impact of urban and rural areas, gender and other aspects.